# Biomass burning induced surface darkening and its impact on regional meteorology in eastern China

Rong Tang[1,2], Xin Huang[1,2,*], Derong Zhou[1,2], and Aijun Ding[1,2]

[1]School of Atmospheric Sciences, Nanjing University, Nanjing, 210023, China
[2]Collaborative Innovation Center of Climate Change, Jiangsu Province, Nanjing, 210023, China

*Correspondence to*: Xin Huang (xinhuang@nju.edu.cn)

**Abstract.** Biomass burning has attracted great concerns for the emission of particular matters and trace gases, which substantially impacts air quality, human health and climate change. Meanwhile, large areas of dark char, carbon residue produced in incomplete combustion, can stick to the surface over fire prone areas after open burning, leading to a sharp drop in surface albedo, so-called 'surface darkening'. However, exploration into such surface albedo declines and the radiative and meteorological effects is still fairly limited. As one of the most high-yield agricultural areas, eastern China features intensive straw burning every early summer, the harvest season for winter wheat, which was particularly strong in 2012. Satellite retrievals show that the surface albedo decline over fire prone areas was significant, especially in the near-infrared band, which can reach -0.16. Observational evidences of abnormal surface warming were found by comparing radiosonde and reanalysis data. Most sites around intensive burned scars show a positive deviation, extending especially in the downwind area. Comparisons between 'pre-fire' and 'post-fire' from 2007 to 2015 indicated a larger temperature bias of the forecast during 'post-fire' stage. The signal becomes more apparent between 14:00 and 20:00 LT. WRF-Chem simulations suggest that including 'surface darkening' can decrease model bias and well captured temperature variation after burning at sites in fire areas and its adjacent area. This work highlights the importance of biomass burning induced albedo change in weather forecast and regional climate.

## 1 Introduction

Biomass burning (BB) refers to open or quasi-open combustion of plants or organic fuels, including forest fires, savannah fires, peat burning and crop residue burning in field (Andreae, 1991; Andreae and Merlet, 2001; Fearnside, 2000; Thompson, 1996). Intensive BB events occur across the world every year, and pose great threats to life and property of human being (Bowman and Panton, 1994; Duncan et al., 2003; Kaiser et al., 2012; Streets et al., 2003; Uhl and Kauffman, 1990) . Apart from this, BB has received widespread attentions due to air pollution, health threats and climate change caused by the large amount of particulate matter, trace gases, and greenhouse gases it releases (Chan and Yao, 2008; Chang and Song, 2010; Chen et al., 2017; Crutzen and Andreae, 1990; Crutzen et al., 1979; Hansen et al., 2005; Haywood and Ramaswamy, 1998; Hobbs et al., 1997; Langmann et al., 2009; Podgorny et al., 2003). However, the impact of BB is much more than that. BB can directly

destruct vegetation covers and affect soil properties over fire prone areas (Certini, 2005), and is considered as one of the most important factors influencing the global terrestrial ecosystem (Fearnside, 2000).

Since open BB tends to occur directly on land surface, vegetation covers will be destructed, and the underlying surface properties would be affected straightway (Certini, 2005; Myhre et al., 2005). Among them, surface albedo is highly sensitive to BB (Jin and Roy, 2005). Due to the cover of black charcoal that produced in incomplete combustion, sharp surface albedo decline can be observed over fire prone areas, which has been proven on several kinds of vegetation covers including savannah and forest (Govaerts et al., 2002; Jin and Roy, 2005; Myhre et al., 2005; Roy and Landmann, 2005). Such kind of 'surface darkening' could maintain for about one week or even longer if it is dryer and calmer, before char materials are removed by weathering and vegetation starts to regenerate (Amiro et al., 2006; Beringer et al., 2003; Bremer and Ham, 1999; Trigg and Flasse, 2000; Tsuyuzaki et al., 2008; Veraverbeke et al., 2012). The magnitude of such surface albedo decline depends on fire intensity and can be up to half of pre-fire values (Veraverbeke et al., 2012). Decreased surface albedo enhances the capacity of fire-affected surface to absorb solar radiation under clear sky condition, perturbing the surface energy budget and the process of land-atmosphere interaction, and then modifying local or even regional circulations (Gatebe et al., 2014). These changes in meteorological parameters will be important for weather and air quality forecast (Ding et al., 2013a; Yang et al., 2019; Zhang et al., 2016).

Though restriction has been strengthened, severe and large-scale BB events still occurred frequently all over the world in recent years (Abatzoglou and Williams, 2016; Nolan et al., 2020). Moreover, threats from global fire activities are still increasing dramatically with climatic and anthropogenic drivers (Marlon et al., 2009; Pechony and Shindell, 2010; Van Der Werf et al., 2006). These facts raise the alarm of further investigation into the impact of BB, to improve human ability to predict future weather and climate (Hansen et al., 2005). Globally, BB events occur with significant seasonal variations and distinct regional characteristics (Duncan et al., 2003; Hao and Liu, 1994; Kaiser et al., 2012; Streets et al., 2003; Uhl and Kauffman, 1990). It is noteworthy that agricultural burning is still very common in many developing countries, mainly during post-harvest periods, for the purpose of clearing farmland and providing ash fertilization for the crop rotation (Gao, 2002; Ravindra et al., 2019; Tipayarom and Kim Oanh, 2020). However, exploration into such fire induced decreased surface albedo and its radiative and meteorological effects is still fairly limited.

As for China, with the world's top-ranked crop production across the world, about 122 Tg of crop residue is burned on spot annually (Yan et al., 2006). Eastern China, especially the middle and lower reaches of Yellow River and Yangtze River, features large plain area and dense population with high agricultural production (Li et al., 2016). Eastern China has been characterized by intensive straw burning every June, the harvest season for winter wheat (He et al., 2007; Huang et al., 2012a, 2018; Li et al., 2016; Yang et al., 2013; Yin et al., 2019). After revolution in rural fuel structure, farmers, who are eager to deal with tons of wheat straw, always resort to burning on spot rather than taking them as fuel (Lu et al., 2011; Yan et al., 2006; Zhu et al., 2019). Hence, the wheat straw burning is exceedingly dense in both time and space (Huang et al., 2012b; Yang et al., 2008). This phenomenon would not only pose alarming threats to air quality and human health (Ding et al., 2016; Nie et

al., 2015), but also bring great challenges to weather and climate forecast (Ding et al., 2013b; Huang et al., 2016; Wang et al.,
2018; Xu et al., 2018; Zhou et al., 2018).
Based on years of satellite fire counts in eastern China, the most severe straw burning happened in June 2012 (Huang et al.,
2012b). The most concentrated area of burning in eastern China is the Yellow-Huai River area (YHR), which connects the
middle and lower reaches of Yellow River and Yangtze River. YHR is the major wheat producing area in China (Lei and Yuan,
2013). An extremely typical pollution episode was triggered by BB emissions from YHR in June 2012, during which a yellow
haze blanketed the Yangtze River Delta (YRD) (Ding et al., 2013a; Huang et al., 2012b; Xie et al., 2015). The intensive BB
events and the caused heavy pollution put weather forecast in trouble, during which a series of meteorological anomalies were
observed, such as air temperature changes and precipitation redistribution (Ding et al., 2013a; Gao et al., 2018; Huang et al.,
2016). On one hand, these anomalies have been partly attributed to aerosols, particularly absorbing aerosols like black carbon
(BC), considering the interaction between air quality and meteorology (Huang et al., 2016). On the other hand, the fire induced
decreased surface albedo and its radiative and meteorological effects in regional scale should not be ignored (Myhre et al.,
2005). The striking spectral contrast between dark charcoal deposit over burned surface and bright ripe wheat could arouse
distinct physical signals, so called 'surface darkening' (Govaerts et al., 2002; Pereira et al., 1999). The influence of such surface
albedo decline on local and regional energy budget can be significant (Jin and Roy, 2005), especially in summer when solar
radiation is extremely strong. It could be another key issue that needs to be well understood so as to figure out the
meteorological anomalies during the burning season. Since many developing areas are still carrying out large-scale agricultural
burning, better understanding of such fire-induced albedo change and its meteorological effects is of great significance for
weather and climate forecast (Ravindra et al., 2019).
In this study, signals of surface albedo decline induced by crop residue burning in eastern China were found based on satellite
retrievals, defined as 'surface darkening'. Magnitude and spatial distribution of the declines were investigated combining fire
counts by MODIS based on straw burning in June 2012. Temperature anomalies after BB were also explored using long-term
site observations from 2007 to 2015 in eastern China, and comparisons were made between the two status, 'pre-fire' and 'post-
fire'. Moreover, with the aid of the Weather Research and Forecasting model coupled with Chemistry (WRF-Chem model),
decreased surface albedo over fire prone areas were taken into consideration in the simulation targeting at temperature
anomalies in June 2012. The further impact of 'surface darkening' in radiation budget, energy reallocation in Earth-atmosphere
system and meteorology were discussed on a local and regional scale, which is of assignable climatological significance and
should not be neglected.

## 2 Data and Method

### 2.1 Fire counts and surface albedo

The intensity of fire counts and the magnitude of surface albedo decline induced by crop residue burning were analysed based on standard Moderate Resolution Imaging Spectroradiometer (MODIS) products provided by NASA (Giglio and Justice, 2015a, 2015b, 2017; Vermote, 2015).

MODIS Surface Reflectance product (MOD09A1) provides an estimate of the surface spectral reflectance of MODIS Band 1 through 7 at a horizontal resolution of 500 m and use Bidirectional Reflectance Distribution Function (BRDF) database to better constrain the different threshold used (Schaaf et al., 2002). Wavelengths of the seven MODIS spectral narrow-bands ranges relatively from 0.62-0.67μm, 0.84-0.87μm, 0.46-0.48μm, 0.54-0.56μm, 1.23-1.25μm, 1.63-1.65μm, and 2.11-2.15μm. The narrowband albedo can be converted to broadband like shortwave and near-infrared through valid algorithms (Liang, 2001, 2003). It has been corrected for atmospheric conditions such as gases, aerosols and Rayleigh scattering. The dataset is updated every 8 d, for selecting one value from all acquisitions in an 8-day composite period for each pixel unit. The criteria for grid cell value selection includes cloud conditions and solar zenith angle.

MODIS Thermal Anomalies/Fire Daily L3 Global product (MOD/MYD14A1) are daily surface thermal anomaly data obtained by Terra and Aqua satellites with spatial resolution of 1 km (Giglio et al., 2009, 2013, 2016; Hawbaker et al., 2008), and fire counts with low confidence were got rid of for quality control. The "fire count" described in the following part indicates the central location of MODIS pixels with surface thermal anomalies. The actual geographic size of the pixel varies depending on the angle at which satellite scans. When counting the number of pixels in target region, the pixels are abstracted into points on a two-dimensional plane.

### 2.2 Temperature observation and forecast

To obtain the surface meteorological conditions, meteorological observations of air temperature at 2 m height were acquired from the Global Hourly Surface Database by the US National Centers for Environmental Information (NCEI). The temperature observations were used to compare with the US National Centers for Environmental Prediction (NCEP) global final analysis data (FNL) with a 1° × 1° spatial resolution which is updated every 6 h, and were also applied for verification and analysis of numerical simulations.

The radiosonde data at Xuzhou (117.15°E, 34.28°N, WMO station number 58027) observatories were collected from the University of Wyoming (http://weather.uwyo.edu/upperair/sounding.html), which is the only sounding site in YHR area. The radiosondes were launched twice a day (08:00 and 20:00 LT) and made measurements both on the mandatory pressure levels (e.g., surface, 1000, 900, 850 hPa) and additional key levels. It measures vertical profiles of atmospheric parameters like air temperature, water mixing ratio, etc.

## 2.3 Experiment design and model configuration

The version 3.6.1 of Weather Research and Forecast Model with Chemistry (WRF-Chem), an online-coupled chemical transport model, was employed in this study, in which multiple physical and chemical processes are depicted including emission, transport, mixing, and chemical transformation of trace gases and aerosols simultaneously with meteorology (Grell et al., 2005). In order to take fire-induced albedo change into consideration and figure out impact of 'surface darkening' on radiation budget and meteorology in eastern China, a pair of parallel experiments in WRF-Chem with surface albedo as variable was designed: In CTL run, default value of monthly surface albedo in model was adopted, while in ABD run, we applied satellite-detected albedo decline over fire-prone areas. In particular, the fire-affected grids were extracted out by certain threshold of fire counts density in June 2012, and the surface albedo decline in each grid is defined as the difference of MODIS-detect surface albedo between 'post-fire' and 'pre-fire' based on satellite data analysis in Section 3.1. In YHR, most fire of wheat straw are characterized by short-lived and incomplete smoldering (Huang et al., 2016). The relationship between the fire radiative power (FRP) and fire sizes varies at a global scale, and FRP of such straw burning is much weaker than grassland fires in North American (Laurent et al., 2019). Based on MODIS Thermal Anomalies Product (MOD/MYD14A1), the average maximum FRP for the most severe burning area in June 2012 is almost less than $0.02 \text{ kW} \cdot m^{-2}$, the energy disturbance of which is much smaller than the continuous radiative disturbance of surface albedo decline in summer. Consequently, the heat released by fire was not treated in the model.

The domain is centered at 115.0°E, 33.0°N with a grid resolution of 20km covered the eastern China and its surrounding areas. The initial and lateral boundary conditions of meteorological variables are obtained from NCEP FNL that update every 6 hour. MIX, a mosaic Asian anthropogenic emission inventory developed by Tsinghua University (Li et al., 2017) and the Fire Inventory from NCAR (FINN) providing high resolution global emission estimates from open burning (Wiedinmyer et al., 2011) were applied in WRF-Chem as anthropogenic and BB emissions, respectively. The simulations were conducted for the time period of 2-18th June 2012, and was split into eight independent 72-hour runs for the meteorological fields, while the chemical outputs from the preceding run were used as the initial conditions for the following run. Only the last 48-hour results of each run were kept for analysis, and the first 7-day were regarded as the model spin-up period for atmospheric chemistry, so as to allow the model to reach a state of statistical equilibrium under the applied forcing (Berge et al., 2001; Lo et al., 2008). The main configurations for WRF-Chem physical and chemical processes in both CTL and ABD experiment are listed in Table 1. A new version of the rapid radiative transfer model (RRTMG) was employed to depict the radiative transfer process for both shortwave and longwave radiation (Iacono et al., 2008; Mlawer et al., 1997). The Noah Land surface scheme (Ek et al., 2003) was used to describe the land-atmosphere interactions, implemented with the MYJ Boundary Layer scheme (Janjic, 1994) to describe the diurnal evolution of the planet boundary layer (PBL). As for cloud and precipitation processes, the new Grell-Freitas cumulus ensemble parameterization (Grell and Freitas, 2014) along with Lin microphysics (Lin et al., 1983) accounting for six forms of hydrometers were employed. Moreover, since the experiments were designed on the basis of

MODIS products, the MODIS surface classification options of the International Geosphere-Biosphere Programme (IGBP) was
adopted to keep consistency.

## 3 Results and discussion

### 3.1 Fire-induced surface albedo change

In 2012, About 160 Tg crop residue was burned, and 36% of the total fires were recorded on farmland, the most of which
occurred in June, the harvest season for wheat (Li et al., 2016). YHR, the major wheat producing area in China, locates in the
vast flat plain in eastern China. Figure 1a shows the daily number of satellite fire counts in YHR, where cropland especially
wheat dominates the vegetation cover (Gong et al., 2019; Lei and Yuan, 2013; Van Der Werf et al., 2010). The limited duration
of agricultural fire combined with the low temporal resolution of MODIS may increase the difficulty of signal capture and
lead to underestimation of fire counts (Chang and Song, 2009, 2010; Giglio et al., 2009; Randerson et al., 2012; Yin et al.,
2019). As illustrated in Fig. 1a, severe and intensive BB events happened in mid-early June, especially 7-13 June, a tense and
busy period in the crop cycle for harvest of wheat and sowing of maize. Among them, burning on 13 June is the most serious.
Black charcoal deposit produced in high-intensive open straw burning would adhere to exposed soil and bound surface albedo
over a period of time before removal of char by natural process or regeneration of vegetation (Veraverbeke et al., 2012). The
strong spectral contrast between dark burned surface and original bright dry ripe wheat cover can arouse distinct physical
signals (Govaerts et al., 2002; Pereira et al., 1999).
Understanding the saliency and persistence of signals is important for estimation of albedo change amplitude and further
analysis of its radiative and meteorological effects. The signals of 'surface darkening' was a relatively short duration for about
one week and much more sensitive in shortwave near-infrared bands (Trigg and Flasse, 2000). Considering higher resolution
and better depiction of narrow-band satellite observations, and the outstanding capacity of near-infrared to separate the signals
between vegetation and charcoal (Jin and Roy, 2005; Trigg and Flasse, 2000), the shortwave near-infrared band (0.84-0.87
μm) was chosen. Figure 1b and 1c show the surface albedo distribution before and after the severe burning separately, based
on MODIS Surface Reflectance product (MOD09A1) which updates every 8 day and has ruled out the interference of clouds.
Data on 1 June has not been adopted because of too few valid data. For shortwave near-infrared band, decrease induced by fire
is about 0.30 to 0.14, almost half of original values. The immediate surface albedo decline after fire can be attributed to the
large-scale replacement of ripe wheat with black charcoal over fire prone areas, so-called 'burned scar'. Char materials strongly
absorb the incoming solar radiation, causing a significant decline of the reflection-to-incoming solar radiation ratio
(Veraverbeke et al., 2012), and this could disturb surface radiation balance at a level that cannot be ignored (Jin and Roy,
2005), which needs further exploration.
The total satellite-detected fire counts with high-confidence from 24 May to 17 June, the highly intensive straw-burning period,
is shown in Fig. 2a. Combined, satellite derived surface albedo changes of the same time period in shortwave near-infrared
broadband at fire locations are shown in Fig. 2b. As illustrated, the fire counts are fairly concentrated, especially in the northern

part of Anhui (AH) province, an agricultural area mainly dominated by winter wheat (Yan et al., 2006). Correspondingly, extensive negative values in Fig. 2b demonstrate substantial signals of surface albedo decline over fire prone area in June 2012. For near-infrared broadband, the declines range from -0.16 to -0.02. In addition, the surface albedo declines show obvious spatial heterogeneity, and have a larger decline margin in northern AH than Jiangsu (JS). The effect of fire surface albedo is complex, determined by combustion completeness, fire intensity, pre-fire land cover structure and underlying soil reflectance (Roy and Landmann, 2005). Here, the burning in AH was indeed more severe, which is consistent with the satellite fire detections. Though the surface albedo product applied here has ruled out cloud interference, partial inevitable noises like cloud shadow still exist, reflecting on the scattering positive values in the north-western region in Fig. 2b.

To better understand the signals agricultural fire induced surface albedo decline and provide a basis for the following numerical experiments, frequency distributions of surface albedo decline in near-infrared (0.7-2.5μm) and short-wave (0.25-2.5μm) band are shown in Fig. 3, to clarify the magnitude of surface albedo decline. The sample size of fire counts is 9,477 pixels, consistent with the sample of those marked in Fig. 2a. Surface albedo change in most fire-affected pixels changes distribute in negative region, while the noises in positive region correspond to those positive values in the north-western area in Fig. 2b owing to cloud shadows. The frequency distribution of both near-infrared and shortwave band shows two apexes, which is in accordance with the spatial heterogeneity of fire in AH and JS. Peak values for shortwave distribute between -0.02 and -0.06, for near-infrared distribute between -0.06 and -0.10. Since nearly half of the solar energy that reaches the surface is at wavelengths longer than 0.7 μm, albedo change in near-infrared is rather significant for the energy budget of the surface (Hartmann, 1994). Consequently, to better characterize the surface albedo decline in aspects of both spatial distribution and scope in severely burned area, the difference of satellite derived surface albedo in shortwave near-infrared band was adopted as the albedo decline between the parallels experiments over fire prone areas.

## 3.2 Observational evidences on surface warming in 'post-fire' period

Surface albedo is defined as the fraction of the downward solar flux density that is reflected by the surface, directly determining the absorptivity of the surface (Hartmann, 1994). In other words, when the flux of downward solar radiation reaching at the surface is the same, more solar radiation will be absorbed by surface with decreased albedo. Without green plants to do photosynthesis over fire prone areas, the immediate fire-induced increase in radiation absorption are used to heat the ground mostly and then warm the near surface atmosphere (Andrews, 2010; Wallace and Hobbs, 2006), leading to the signal of surface warming. What's more, straw burning in YHR bursts in summer, when solar radiation is the strongest and the response of surface albedo can be the fiercest. Hence, the air temperature observations over fire prone areas after BB were investigated.

Near-surface air temperature observations at meteorological sites were compared with FNL data in 2012. Refer to Fig. 1a, burning in YHR is the most severe on 13 June, while 18 June is the end date of the wheat straw burning 'season' in 2012. Temperature bias at 2 m height in observational network at 20:00 LT on 13 and 18 June was shown in Fig. 4, filled according to the value of observation minus FNL. In addition, fire counts on the day and accumulated during the past five days, so called 'burned scar', were marked. The scattered positive values of temperature bias in Fig. 4 represent that observational temperature

at those sites are higher than predictions from forecast model with only a small fraction of assimilated observations. Most sites
around intensive burned scars shows a positive deviation, which is an abnormal signal of surface warming, extending especially
in the downwind direction.
In fact, intensive straw burning occurred in YHR every June, during post-harvest period of wheat over the past twenty years
before the launched strict regulations in recent years (Li et al., 2016; Yin et al., 2019), which could announce significance in
climatology to understand the signals of surface warming after BB.
Xuzhou (XZ) station (marked in Fig. 4), the only radiosonde observatory located in the most intensive fire prone area in YHR.
The radiosonde observations at XZ from 2007 to 2015 was compared with FNL data in the PBL. Sample of days was selected
out according to burning condition in each year, and has been classified into specific status, 'pre-fire' and 'post-fire'. 'Post-
fire' days must be in 5 days after a severe burning, while 'pre-fire' must be among the 5 days before fire but not in 5 days after
previous fire. Only days under clear sky condition were selected. The statistical distribution of temperature bias on different
isobaric levels at 20:00 LT in the two status was shown in Fig. 5a and 5b. Temperature bias was defined as the value of
radiosonde observation minus FNL data. As illustrated, temperature biases are mostly positive on the lower levels in both two
status, and absolute values of 'post-fire' are larger than 'pre-fire'. Combined with Fig. 5c, it means observations at 'post-fire'
have bigger positive deviation from forecast results, a signal of abnormal warming at 'post-fire'. In addition, with regards to
the vertical profile of bias at 'post-fire' (Fig. 5b), deviations on lower levels tend to be bigger and more positive than those on
higher levels. Larger absolute value of deviation means lower predictability of temperature at 'post-fire'. Radiosonde data
updates 2 times a day, 08:00 LT and 20:00 LT, and only patterns at 20:00 LT was shown here since no significant difference
exists between the two status at 08:00 LT. After accumulation of absorbed radiation during the daytime, the change of thermal
states near the ground level can be notable by 20:00 LT. The abnormal warming signal tends to be more obvious near the
surface, which can be explained by the enhancement of surface heat absorptivity caused by decreased surface albedo.
For the same sample of days in 2007-2015, the diurnal variation of 2-m temperature bias between site observations and FNL,
in the two status was shown in Fig. 6. Negative values of deviation at 08:00 LT and 14:00 LT show that FNL tends to
overestimate temperature during daytime and underestimate at night. It is noteworthy that the absolute values of deviations at
'post-fire' are always larger than those at 'pre-fire' no matter during the day or night. With regard to the median values,
deviations at 'post-fire' gradually surpass over those at 'pre-fire' between 14:00 and 20:00, consistent with the warming signal
revealed by radiosonde analysis at 20:00 LT. The continuous uptrend during the daytime demonstrates a needed response time
of heat accumulation, which may be attributed to the radiative effect of 'surface darkening' (Andrews, 2010;Myhre et al.,
2005). Thus, the signal of abnormal warming becomes more apparent between 14:00 and 20:00 LT, with the gradual
accumulation process.
**3.3 Improved temperature simulation by considering surface albedo change**
As aforementioned, fire-induced surface albedo change and observational evidences of abnormal surface warming were
substantial in eastern China. However, the radiative effects on meteorology and the underlying physical images of surface

warming have not been figured out yet, which should resort to model simulation. In existing studies, radiative effects of albedo changes over burned scars were investigated on land covers, like boreal forests (Lyons et al., 2008) and savannas (Jin and Roy, 2005; Myhre et al., 2005), but few of them was conducted on open agricultural fire in farmlands. Exploration into the disturbance of surface albedo decline induced by agricultural BB using model simulation can be meaningful for improving weather and climate forecast of regions with the large scale BB events. Hence, WRF-Chem experiments targeting at surface albedo change induced by agricultural BB in eastern China were conducted. The most severe episode happened in June 2012 was selected as the case. Surface albedo set in the two parallel experiments, CTL and ABD, were shown in Fig. 7a and 7b. Figure 7c shows the surface albedo difference between the two experiments, which is defined as the difference of MODIS-detect surface albedo between 'post-fire' (17 June) and 'pre-fire' (24 May) in each grid. Model results were compared with site observational air temperature, respectively in concentrated fire prone areas (Fig. 8a) and its southern adjacent area, YRD (Fig. 8b). In Fig. 8, ABD experiment shows increase in temperature and enjoy a better fit with observations at both sites in the afternoon and evening compared to CTL experiment. Temperature anomalies can be simulated to some extent, after considering the decreased surface albedo caused by BB.

The local surface energy budget at Bengbu was shown in Fig. 9. Bengbu locates in fire affected area where adopted the decreased albedo. As shown in Fig. 9a, upward shortwave radiation at the surface reduces a lot, while only a little change exists in downward shortwave radiation reaching the surface. The smaller ratio of solar radiation reflected to the atmosphere by surface can be attributed to surface albedo decline. Accordingly, with more solar radiation absorbed by the surface, the ground will be heated and the surface temperature will be higher in ABD experiment (Fig. 9b). Higher surface temperature would give rise to larger upward heat flux, thereby heating the atmosphere near the surface by vertical mixing (Stull, 1988). So, local radiation budget change induced by surface albedo change would directly influence the near-surface air temperature and arouse an abnormal warming signal over fire prone areas in the afternoon and evening with a certain lag.

Meanwhile, the temperature influence of fire-induced albedo change is also obvious in adjacent areas like LK (Lukou) (Fig. 8b), located in YRD. Some other stations in YRD also show similar patterns. Local temperature can be affected by various factors such as local radiation budget change and advection transport caused by thermal disturbance. As illustrated in Fig. 10a, surface warming shows in the upwind area of LK station, which means influence of warm advection. According to Fig. 10b and 10c, sharp decrease in low cloud water contents was found in the upwind area in ABD experiment. Low clouds, effective reflectors of solar radiation (Hartmann, 1994), would arouse large perturbations to surface radiation budget. In Fig. 10d, downward shortwave reaching the surface in ABD experiment reduces quit a lot due to reflection and occlusion of low clouds. The additional heat by decreased albedo over fire prone areas would increase the surface temperature, change both sensible heat flux and latent heat flux, and then change the vertical velocity in boundary layer and even disturb the process of cloud formation (Stull, 1988). Thus, surface albedo declines near fire locations also have great impacts on areas where surface albedo was barely changed, by disturbance to cloud formation and advection transmission.

These 'burned scars' can remain over the surface for about one week under rainy condition, or even longer if dryer and calmer
(Trigg and Flasse, 2000). The length of period allows the process of local radiative accumulation and disturbance in regional
circulation, and influence on temperature over both fire prone areas and adjacent areas.
**4 Conclusions**
To figure out surface albedo change induced by biomass burning and its impact on regional meteorology in eastern China, an
investigation into the relationship between surface albedo change and temperature anomalies was conducted based on
meteorological observations and satellite retrievals, combining with WRF-Chem simulations. This study focuses on the
intensive wheat straw burning occurred every early summer in the YHR area, the major wheat producing area in eastern China.
A typical and severe burning episode in 2012 was chosen to be the study case.
Fire-induced 'surface darkening' over fire prone areas was verified based on satellite retrievals. Large area of surface albedo
decline shows over fire prone areas in YHR. It also shows obvious spatial heterogeneity, and has a larger decline margin in
AH, where fire counts are more concentrated and the burning is much more severe. For the near-infrared broadband, the
absolute surface albedo decline ranges in -0.16 ~ -0.02, which can be an obvious signal in the regional scale and arouse large
radiative disturbance. Peak values of reduction for shortwave band (0.7-2.5μm) distributed between -0.06 and -0.02, for near-
infrared band (0.25-2.5μm) distribute between -0.10 and -0.06.
Evidences of abnormal surface warming were found in eastern China. Most sites around intensive burned scars shows a positive
deviation in June 2012, an abnormal signal of surface warming which extends especially in the downwind direction.
Comparisons were made between status of 'pre-fire' and 'post-fire' under clear sky condition, based on multi-years
temperature observations from 2007 to 2015. Observations at 'post-fire' have bigger positive deviation from forecast results
at 20:00LT, especially on lower levels. This kind of abnormal surface warming signal at 'post-fire' will lead to lower
predictability of temperature. In terms of diurnal variation, the signal becomes more apparent between 14:00 and 20:00 LT,
with the accumulation of absorbed radiation.
To examine the direct radiative effects and its potential regional meteorological impacts of surface albedo change, WRF-Chem
experiments targeting at the surface albedo change induced by agricultural BB in eastern China were carried out. ABD
experiment shows increase in temperature and enjoys a better fit with observations in concentrated fire prone areas and its
southern adjacent area compared to CTL, especially in the afternoon and evening. Surface albedo change over fire prone areas
influence the surface temperature through both direct local radiation budget change and indirect disturbance in cloud formation
and advection transmission.
This study shows that either human or naturally induced biomass burning will not only influence the weather and regional
climate by emission of aerosols and trace gases. The change in surface albedo could also cause significant impacts at regional
scale during the post-fire periods, particular in regions with strong solar radiation. Such kind of short-term disturbance should
be considered in future weather, climate and air quality forecast models.
*Data availability.* Meteorological datasets used in this work can be acquired from https://rda.ucar.edu/datasets/ds083.2/,
http://weather.uwyo.edu/upperair/sounding.html, and https://www7.ncdc.noaa.gov/CDO/cdoselect.cmd. MODIS dataset can
be obtained from the Level-1 and Atmosphere Archive & Distribution System (LAADS) Distributed Active Archive Center
(DAAC), located in the Goddard Space Flight Center in Greenbelt, Maryland (https://ladsweb.nascom.nasa.gov).
*Author contributions.* RT and XH led the manuscript writing. RT and DZ conducted the data analysis and model simulations.
AD contributed to the research design.
*Competing interests.* The authors declare that they have no conflict of interest.
*Acknowledgements.* This work was supported by the National Natural Science Foundation of China (41922038, 41725020,
and 91744311). The numerical modeling was conducted on computing facilities at the High Performance Computing Centering
(HPCC) at Nanjing University.

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

**Table 1. WRF-Chem domain setting and configuration selections.**

| Domain setting | |
| --- | --- |
| Horizontal grids | $130 \times 130$ |
| Grid spacing | $20km \times 20km$ |
| Vertical layers | 30 |
| Map projection | Lambert Conformal |
| **Configuration selections** | |
| Land surface | Noah |
| Boundary layer | MYJ |
| Microphysics | Lin et al. |
| Cumulus | Grell-Freitas |
| Radiation | RRTMG |
| Chemistry | CBMZ |
| Aerosol | MOSAIC |


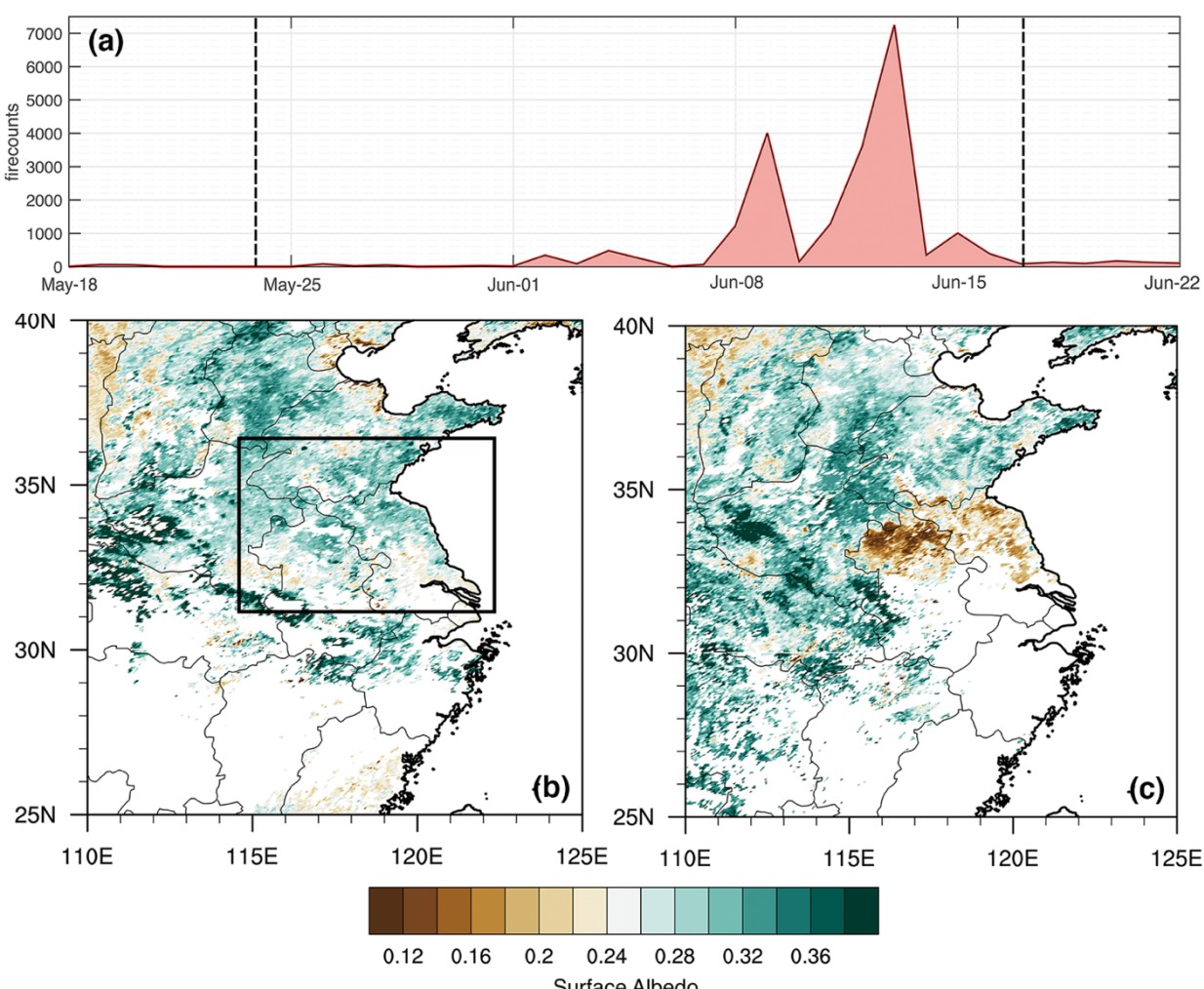


Figure 1. (a) Daily number of satellite fire counts in the black-boxed zone marked in Fig. 1b. Distribution of satellite-retrieved surface albedo in the northern part of Anhui and Jiangsu Province (b) on 24 May, (c) on 17 June in 2012. Note that the two dates are marked by black dashed lines in Fig. 1a.

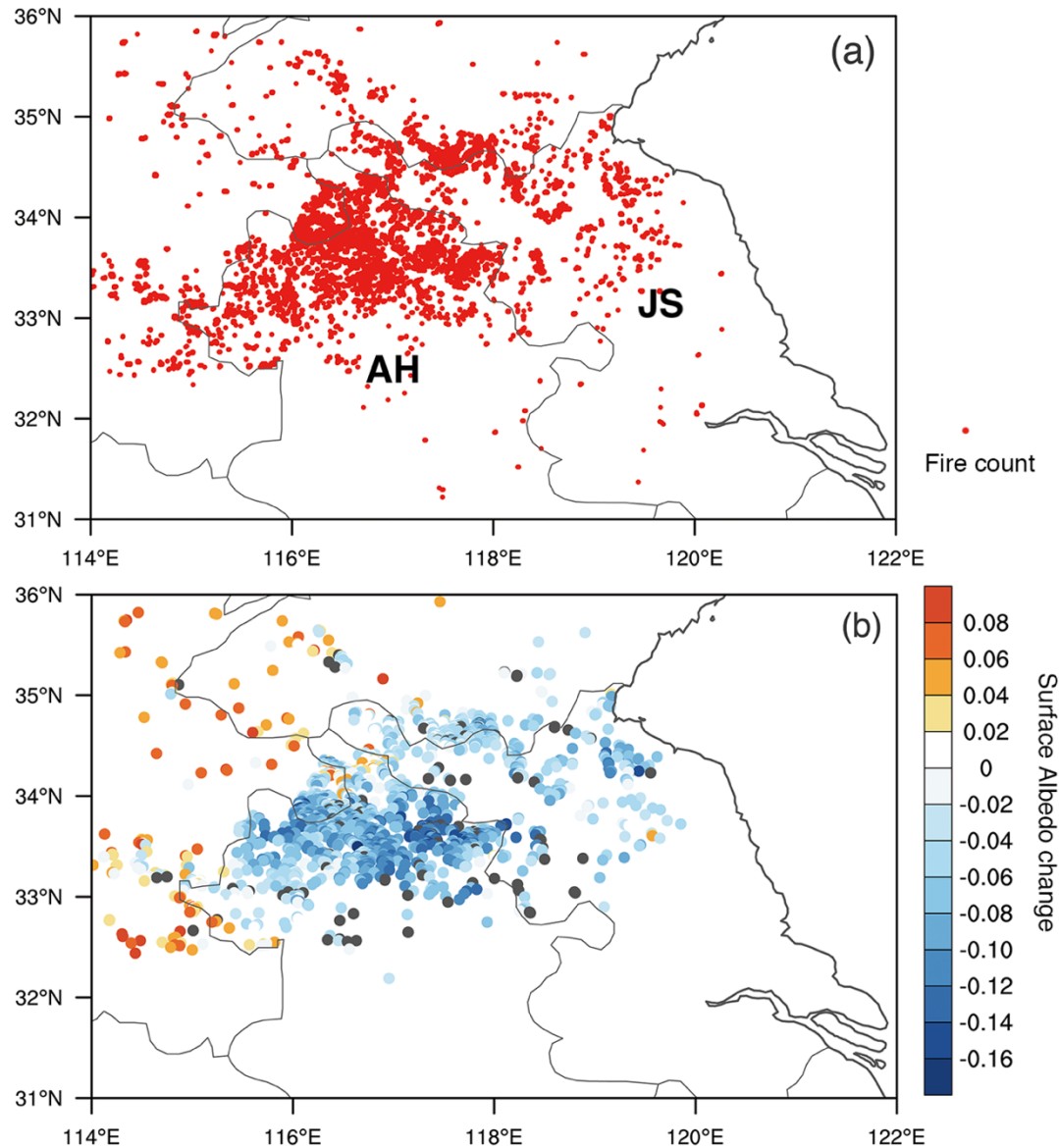

**Figure 2. (a)** Accumulated satellite fire counts with high-confidence from 24 May to 17 June in 2012. **(b)** Surface albedo changes for the period by MOD09A1 over corresponding burned area in Fig. 2a. Declines equal values on 17 June minus those on 24 May. Note that a certain uniform data interval was adopted considering the dots' density.

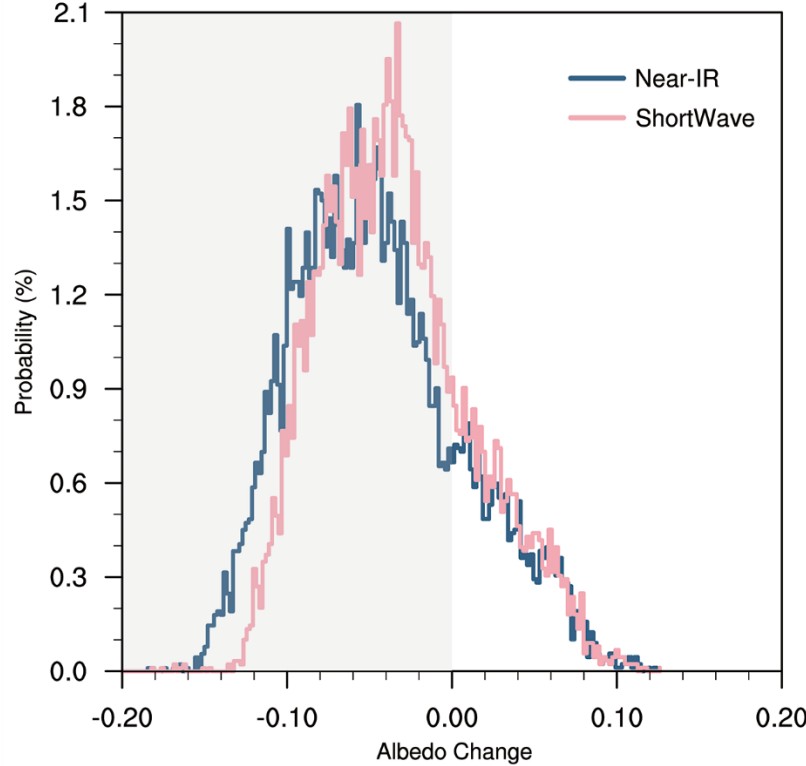

44

**Figure 3.** Frequency distributions of satellite-retrieved surface albedo changes in all pixels with high-confident satellite fire counts. Changes equal values on 17 June minus those on 24 May. The wavelength ranges of 'Near-IR' and 'Shortwave' are '0.7-2.5$\mu m$' and '0.25-2.5 $\mu m$', respectively.

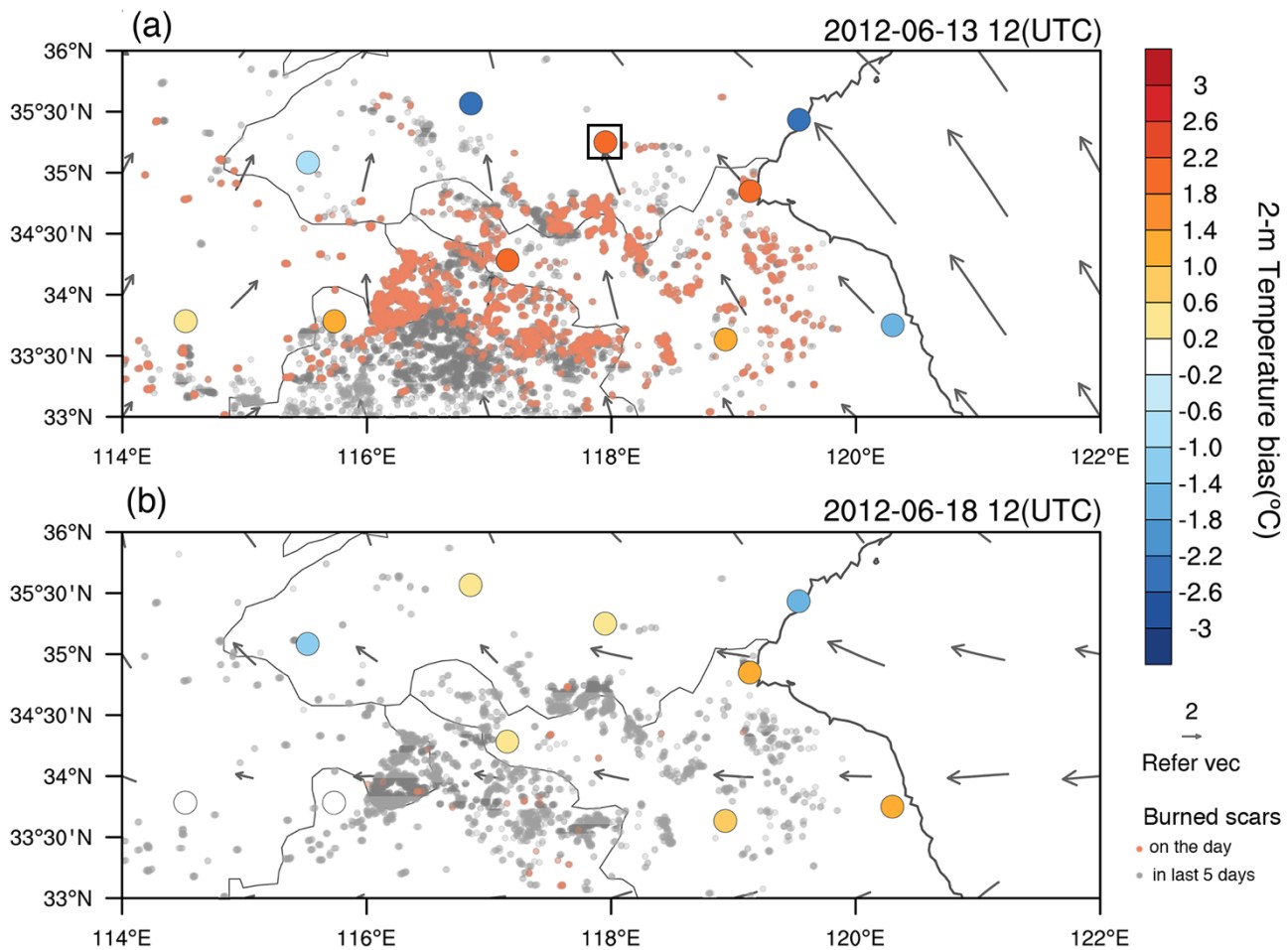

**Figure 4.** Temperature bias at 2 m height (fill-coloured circles) between FNL analysis data and station observations (OBS) in zone YHR **(a)** on 13 June 2012 and **(b)** on 18 June 2012, at local time 20:00. Bias is defined as value of OBS minus FNL. Grey arrows mark the 10-m wind field in FNL. Grey dots mark the 'burned scars', which are defined as accumulated fire counts in the past 5 days. Orange dots mark fire counts on the day. Note that the small black box in Fig. 4a marks the location of XZ (Xuzhou).

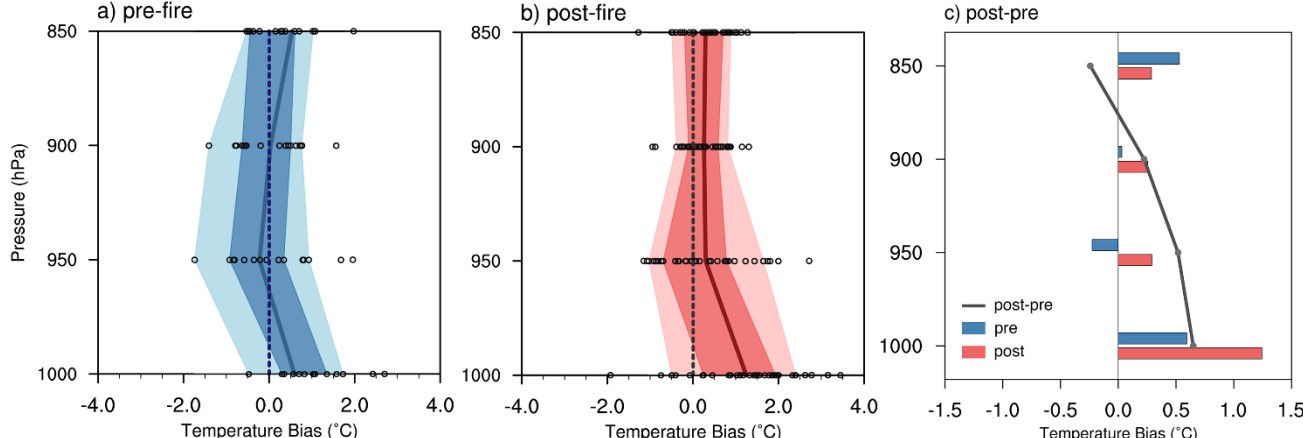

**Figure 5.** Temperature Bias on isobaric levels between FNL analysis data and radiosonde observations (OBS) at local time 20:00 at XZ (Xuzhou) on clear-sky days before and after BB in June 2007-2015: **(a)** pre-fire **(b)** post-fire. Black circles mark original value of temperature bias (OBS-FNL), dash line marks zero, and the five curves of filled parts are respectively 10, 25, 75, 90 percentile and average line in proper order. Bars in **(c)** are the mean value at separate levels and the curve is the corresponding bias between 'post-fire' and 'pre-fire'. Note that the small black box in Fig. 4a marks the location of XZ (Xuzhou).

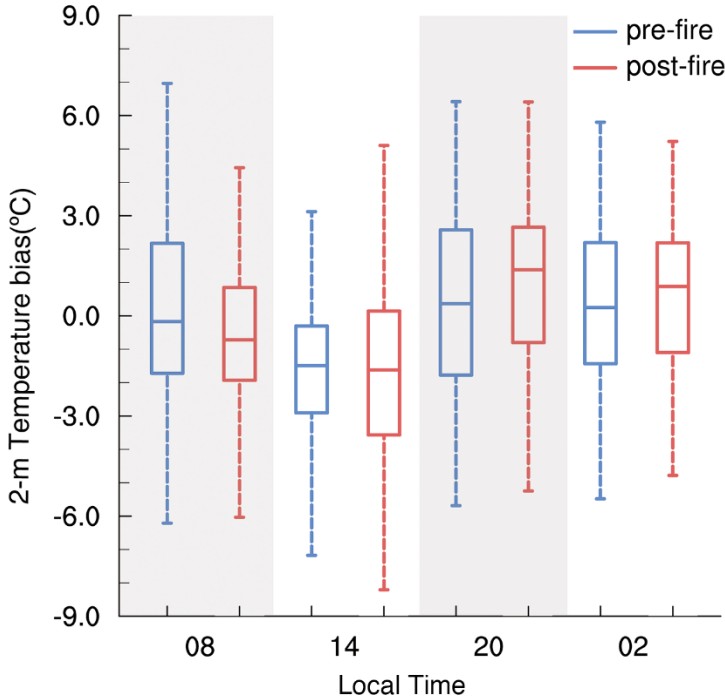

**Figure 6.** Temperature bias at 2 m height, defined as the value of OBS minus FNL, at stations over fire prone area at local time 08:00, 14:00, 20:00, 02:00 under clear-sky condition pre/post fire in 2007-2015.

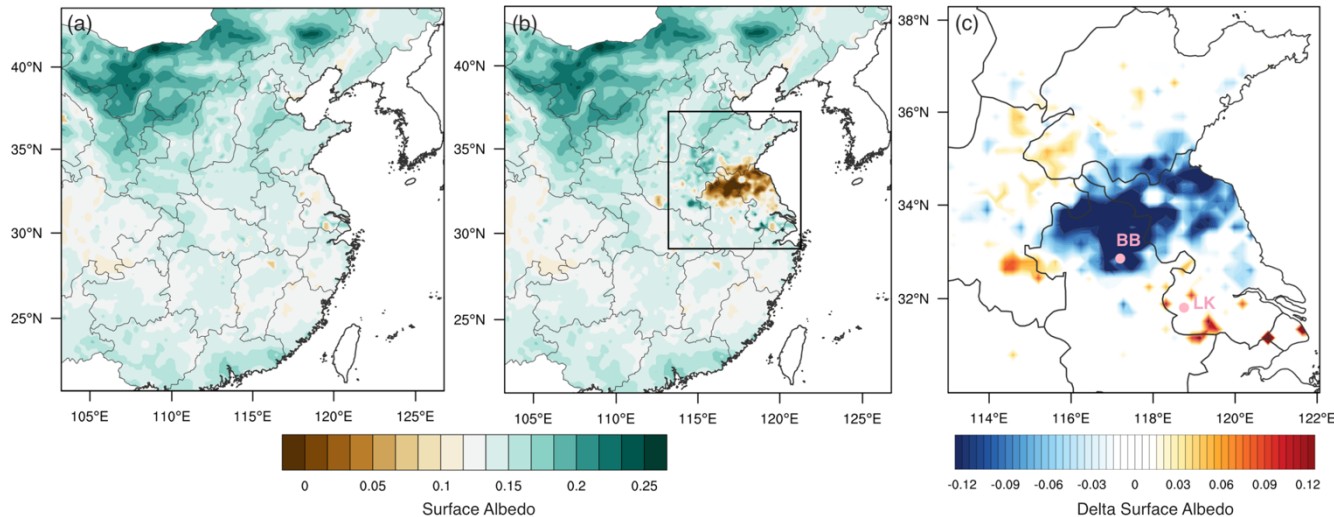

**Figure 7.** Surface albedo distribution in experiments: **(a)** CTL and **(b)** ABD, and **(c)** difference between the parallel ones (ABD-CTL). Note that region in Fig. 7c corresponds to the black box in Fig. 7b.

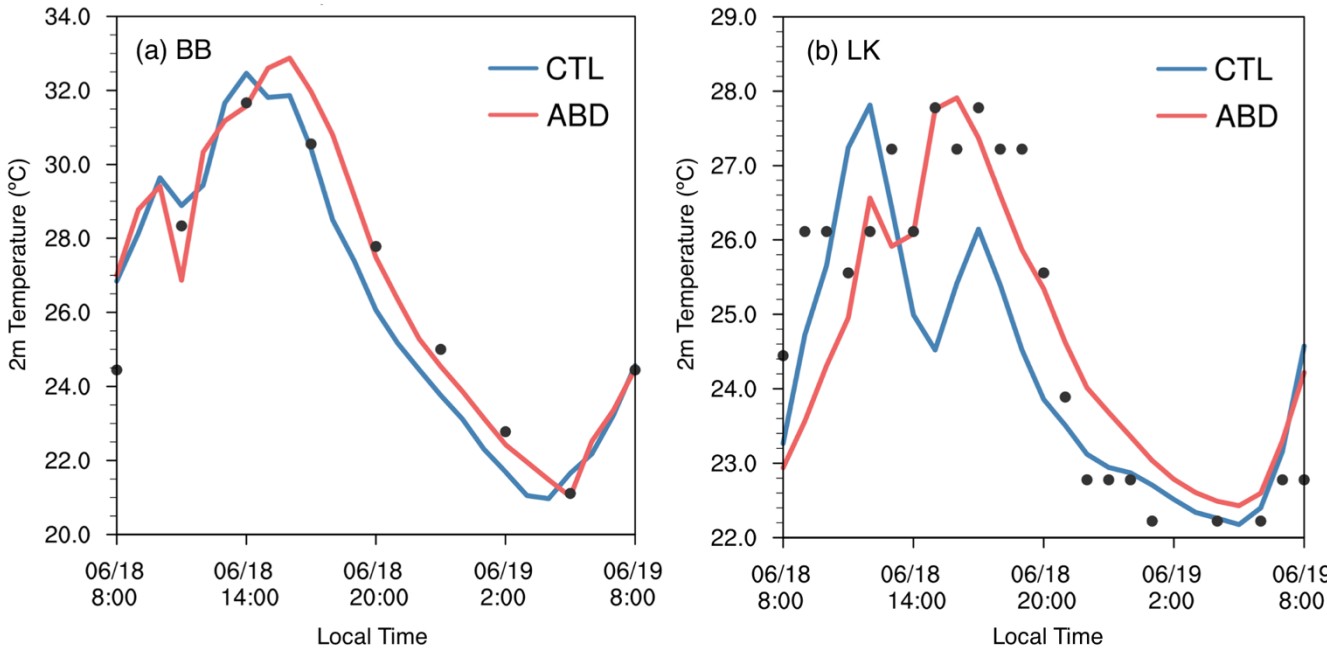

**Figure 8.** Simulated and observed 2-m temperature at meteorological stations: **(a)** BB (Bengbu), **(b)** LK (Lukou), on 18 June 2012. Note that locations of BB (Bengbu) and LK (Lukou) are marked in Fig. 7c.

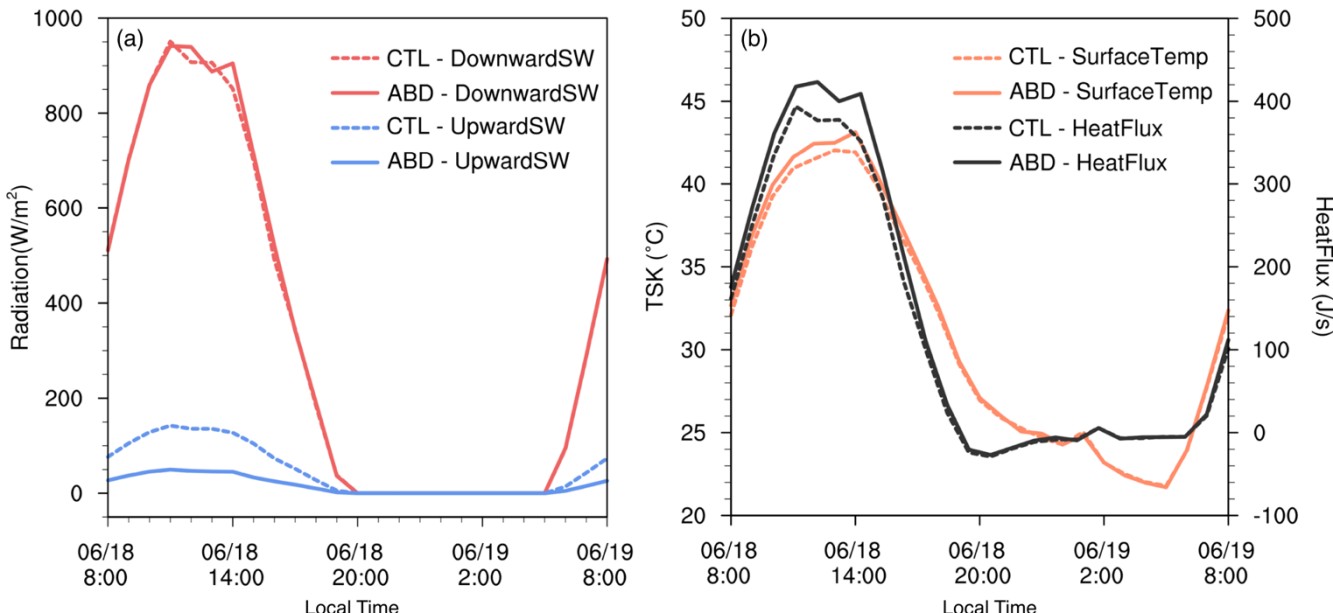

**Figure 9.** Thermal physical quantities at surface level on 18 June 2012 in experiments CTL and ABD. **(a)** Down-ward and upward shortwave radiation, **(b)** Surface temperature and upward heat flux.

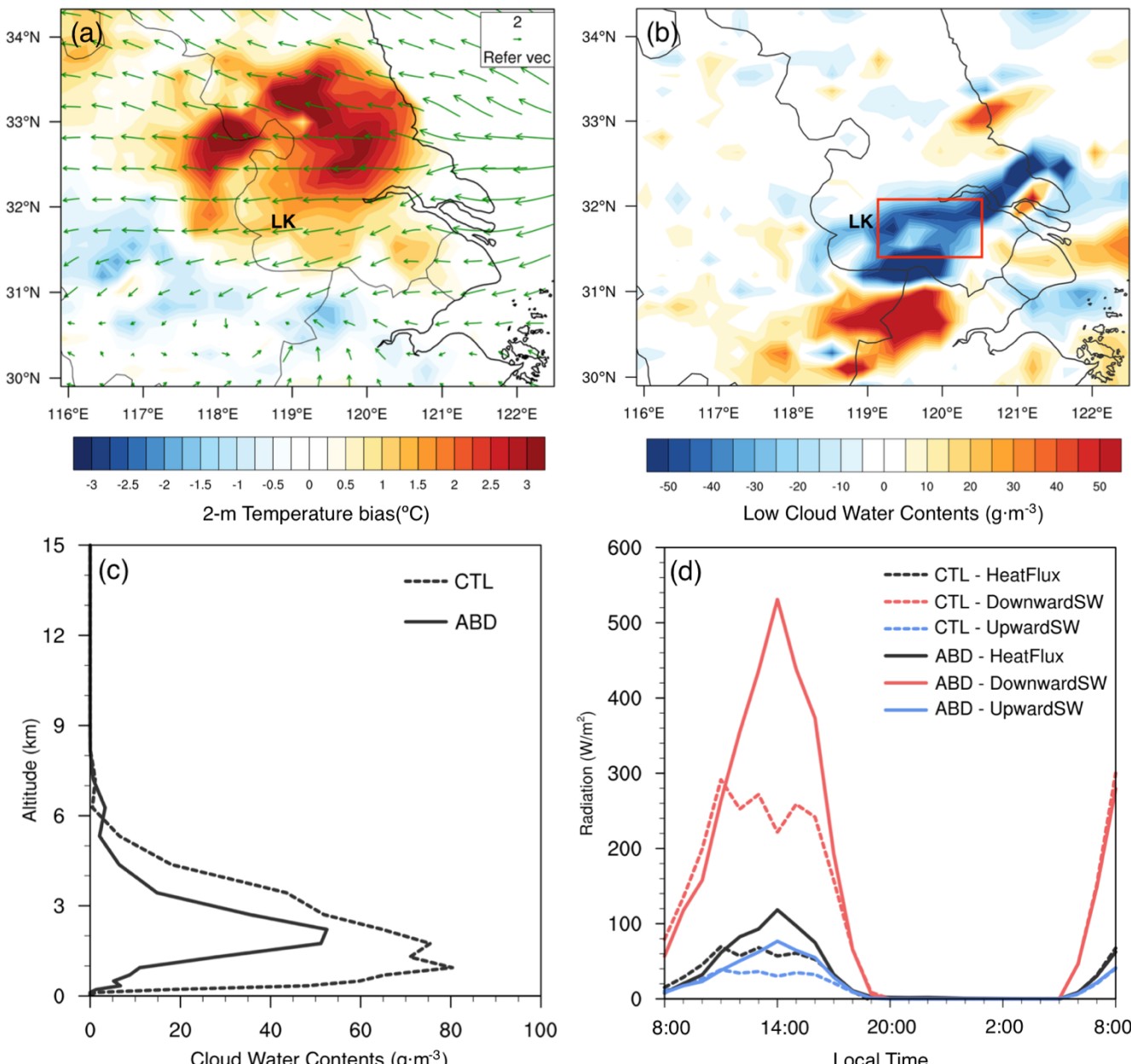

**Figure 10. (a)** Average 10m-wind field in ABD and difference of 2m-temperature between ABD and CTL (ABD-CTL) during the afternoon (12-17LT) on 18 June. **(b)** Average difference in low cloud water contents between ABD and CTL (ABD-CTL). **(c)** Vertical profile of cloud water content during afternoon (12-17LT) and **(d)** shortwave radiation and upward heat flux averaged in region marked by red rectangle in Fig. 10b on 18 June.