# Peer review of "Biomass burning induced surface darkening and its impact on"

_Atmospheric Chemistry and Physics, 2019_

## Referee Comment (RC1) · Anonymous Referee #1 · 1 Jan 2020

This manuscript explores how biomass burning impacts surface albedo and regional meteorology using radiosonde, satellite retrievals, reanalysis data and the WRF-Chem model. It is well organized and written. The results demonstrate improvements in modeling meteorology during biomass burning seasons. It is worth to be published in ACP after addressing the following issues.

My major concern is on the treatments of changes in surface albedo in the model. Observed changes in albedo for shortwave and near infrared are -0.02∼-0.06, and -0.06∼-0.1. The largest decrease of -0.1 was used in the ABD run, while changes of about -0.05 occurred more frequently. Will this treatment overestimate the impacts of

biomass burning in the model? Could you discuss how this will affect your conclusion.

Page 6, line 190-191: why abnormal signal of surface warming happens in the down-wind direction? Is fire induced surface warming treated in your model?

Minor suggestions

Page 1, line 8: impacts air quality

Page 1, line 13: show that surface

Page 1, line 15: show a positive deviation

Page 4, line 102: meteorological observations from

Page 7, line 221: "and the behind physical images has" to "and the underlying physical images have not "

Page 7, line 222-224: please rephrase this sentence to make it easier to read

Page 9, line 262: change the sentence to "A typical and severe burning episode in 2012 was chosen as the study case."

Page 9, line 279: naturally induced

---

## Referee Comment (RC2) · Anonymous Referee #3 · 28 Feb 2020

The manuscript explored the biomass burning induced surface darkening effects and the climate feedback on regional meteorology in eastern China combined with satellite retrievals and WRF-Chem simulations. The paper fits the scope and deserves to be published in this journal with minor revisions.

Major concerns are listed as follows:

1. Line 12-13: "Satellite retrievals show that surface ......in the near-infrared broadband)." This sentence is unclear. Do you mean the decreased surface albedo (-0.16) appears in the harvest season than other seasons? Please rewrite this sentence.

2. Line 17: The same as above. What do you mean of later afternoon, should be

rewritten more accurately.

3. Line 29-33: The sentence is too long and hard to follow. Please revise to short sentences.

4. Line 37-41: As above, please shorten the sentence.

5. The multi citations in this manuscript should be separated by blank space.

6. Line 42: Grammar mistake for "a decrease". The form"a" is rarely used in front of "decrease". The sentence can be rewritten as "such decreased surface albedo depends..." or "such surface albedo decline depends..."

7. Line 48-49: Suggested to delete "in estimation"

8. Line 50-51: Delete "Owing to......maize,"

9. Line 53-54: The sentence can be revised as "farmers, who are eager to deal with tons of wheat straw, always resort to burning on spot rather than taking them as fuel.

10. Line 56: Please be careful to follow the format of this journal. (Ding et al., 2013a; 2015a, 2015b).

11. Line 69-70: Shorten the sentence.

12. Line 71: Should be revised as "decreased surface albedo or surface albedo decline". similar changes throughout the manuscript.

13. I suggested that the introduction section should be reorganized or revised by native editors.

14. Line 84: (MOD09A1), the other similar changes should also be revised.

15. Figure 1 b-c. The color bar of Figure 1b and 1c should be labeled (units)

16. In Figure 3, the wavelengths for shortwave and near_IR should be given.

---

## Author Comment (AC2) · 29 Mar 2020

**Response to Referee #2**

*General Comments:*

*The manuscript explored the biomass burning induced surface darkening effects and the climate feedback on regional meteorology in eastern China combined with satellite retrievals and WRF-Chem simulations. The paper fits the scope and deserves to be published in this journal with minor revisions.*

**Response:** We appreciate for the professional review on our article. According to the suggestions, we will make extensive corrections to improve the manuscript, the detailed corrections are listed below.

*Specific Comments:*

*1. Line 12-13: "Satellite retrievals show that surface . . .. . .in the near-infrared broadband)." This sentence is unclear. Do you mean the decreased surface albedo (-0.16) appears in the harvest season than other seasons? Please rewrite this sentence.*

**Response:** Thanks for pointing it out. Based on satellite retrievals, sharp surface albedo declines are found over fire prone areas. And the scope of declines in the near-infrared broadband is the most significant, which can be up to -0.16. It is consistent with the outstanding capacity of near-infrared band to separate the signals between vegetation and charcoal, as demonstrated by previous studies (Jin and Roy, 2005; Trigg and Flasse, 2000).

*2. Line 17: The same as above. What do you mean of later afternoon, should be rewritten more accurately.*

**Response:** Thanks for the suggestions. The 'later afternoon' here referred to the time after 14:00 until the sunset. It will be rewritten more accurately in the revised version for clarity.

*3. Line 29-33: The sentence is too long and hard to follow. Please revise to short sentences.*

**Response:** Accepted. The sentence will be rephrased in the revised version.

*4. Line 37-41: As above, please shorten the sentence.*

**Response:** Accepted. The sentence will be shortened in the revised version.

*5. The multi citations in this manuscript should be separated by blank space.*

**Response:** Accepted. All citations in the manuscript will be separated by blank space in the revised version.

*6. Line 42: Grammar mistake for "a decrease". The form "a" is rarely used in front of "decrease". The sentence can be rewritten as "such decreased surface albedo depends..." or "such surface albedo decline depends..."*

**Response:** Accepted. Thank you very much for pointing it out and we will correct it accordingly.

*7. Line 48-49: Suggested to delete "in estimation"*

**Response:** Accepted.

*8. Line 50-51: Delete "Owing to......maize,"*

**Response:** Accepted.

*9. Line 53-54: The sentence can be revised as "farmers, who are eager to deal with tons of wheat straw, always resort to burning on spot rather than taking them as fuel.*

**Response:** Accepted. This revision will be made for clarity.

*10. Line 56: Please be careful to follow the format of this journal. (Ding et al., 2013a; 2015a, 2015b).*

**Response:** Accepted. We were really sorry for our careless mistakes. Article citation format will be carefully checked in the revised version.

*11. Line 69-70: Shorten the sentence.*

**Response:** Accepted. The sentence will be shortened in the revised version.

*12. Line 71: Should be revised as "decreased surface albedo or surface albedo decline". Similar changes throughout the manuscript.*

**Response:** Accepted. Similar expressions will be modified in the revised version.

*13. I suggested that the introduction section should be reorganized or revised by native editors.*

**Response:** Accepted. We will reorganize the introduction section. In addition, more detailed statements of situations in China and relevant previous studies will be added to make the background clearer.

*14. Line 84: (MOD09A1), the other similar changes should also be revised.*

**Response:** Accepted. Similar changes for MODIS datasets will be done in the revised version.

*15. Figure 1 b-c. The color bar of Figure 1b and 1c should be labeled (units)*

**Response:** Accepted. The physical quantity of the color bar is a ratio, 'surface albedo'. The unit and related indication will be added in the revised version.

*16. In Figure 3, the wavelengths for shortwave and near_IR should be given.*

**Response:** Accepted. The sentence will be shortened in the revised version.

**References**

Jin, Y., and Roy, D. P.: Fire-induced albedo change and its radiative forcing at the surface in northern Australia, GEOPHYS RES LETT, 32, 10.1029/2005gl022822, 2005.

Trigg, S., and Flasse, S.: Characterizing the spectral-temporal response of burned savannah using in situ spectroradiometry and infrared thermometry, INT J REMOTE SENS, 21, 3161-3168, 10.1080/01431160050145045, 2000.

---

## Author Response (AR1)

**Response to Referee #1**

*General Comments:*

*This manuscript explores how biomass burning impacts surface albedo and regional meteorology using radiosonde, satellite retrievals, reanalysis data and the WRF-Chem model. It is well organized and written. The results demonstrate improvements in modeling meteorology during biomass burning seasons. It is worth to be published in ACP after addressing the following issues.*

**Response:** We would like to thank the referee for providing the insightful suggestions, which have indeed helped us further improve the manuscript.

*Specific Comments:*

*1. My major concern is on the treatments of changes in surface albedo in the model. Observed changes in albedo for shortwave and near infrared are -0.02∼-0.06, and -0.06∼-0.1. The largest decrease of -0.1 was used in the ABD run, while changes of about -0.05 occurred more frequently. Will this treatment overestimate the impacts of biomass burning in the model? Could you discuss how this will affect your conclusion.*

**Response:** Thanks for the suggestions. Sensitivity tests for different surface albedo decline have been supplemented in the following part. Related discussion have been modified in the revised manuscript.

In this study, we aim to figure out the possible radiative effects of straw burning induced surface albedo decline and its impact on regional meteorology. Many existing studies have indicated that the effect of fire on surface albedo is complex and depends on combustion completeness, fire intensity, pre-fire land cover structure,underlying soil reflectance (Roy and Landmann, 2005). Based on satellite retrievals, the surface albedo declines in June 2012 show obvious spatial heterogeneity (Fig. R1), and have a larger decline margin in Province Anhui (AH) than Jiangsu (JS). It is relatively consistent with the two apexes on the frequency distribution of albedo decline (Fig. 3). And the burning in AH was indeed more severe, which is consistent with the distribution of fire detection by satellite. In the numerical experiments,the fire prone areas were extracted out by setting a threshold number of fire counts in a grid unit.

To further understand the impact of the albedo decline values on the conclusions, three experiments are supplemented: On the one hand, to compare with the ALB-0.1 run (namely the ABD run in manuscript), the decline margin in run 'ALB-0.05' and 'ALB-0.08' are set to -0.05 and -0.08, respectively. One the other hand, decline values in run 'ALB-△modisalb'

were set by the difference of MODIS-detect surface albedo (MOD09A1) on 24 May and 17 June directly. The distribution of albedo change is shown in Fig. R2. It is clearly demonstrated that the declines of most areas in northern AH, featuring the highest fire density, are even far more than 0.12. In addition, albedo changes in adjacent areas are less than 0.06, or have not been extracted as fire prone area for higher threshold of fire counts density here.

Results of these runs and site observations were compared. In Fig. R3a, both 'ALB-0.1' and 'ALB- △ modisalb' perform well at Bengbu, especially in the evening. In contrast, by comparing 'ALB-0.1', 'ALB-0.08' and 'ALB-0.05', we can find reverse increment in the evening, which can be explained by the longwave radiation balance in the evening. As for the surrounding areas with slight burning (Fig. R3b), warmings at noon have a positive increase among 'ALB-0.05', 'ALB-0.08' and 'ALB-0.1'. And the 'ALB- △ modisalb' with the direct albedo difference of pre-fire and post-fire even performs better, owing to the better characterization of surface albedo decline, in aspects of both spatial distribution and scope in severely burned area. Your valuable and thoughtful comment led me to explore more deeply about the heterogeneity of surface albedo change and the albedo change set in sensitivity tests. Related discussion have been   modified in the revised manuscript, and more descriptions on the scope and distribution characteristics of surface albedo decline have been added. Please see line 127-131, 183-192, 199-205, 258-283 and 293-295 in the revised manuscript. Accordingly, Figs. 7-10 have been alternated with new results in ABD experiment.

[Figure]

**Fig. R1** Distribution of surface albedo by MODIS band2 on **(a)** 24 May, **(b)** 17 June in 2012. Two provinces Jiangsu (JS) and Anhui (AH) are marked respectively.

[Figure]

**Fig. R2** Distribution of surface albedo change set in experiment 'ALB- Δ modisalb'.

[Figure]

**Fig. R3** Simulated and observed 2-m temperature at meteorological stations: **(a)** Bengbu and **(b)** Lukou.

*2. Page 6, line 190-191: why abnormal signal of surface warming happens in the down-wind direction? Is fire induced surface warming treated in your model?*

**Response:** Thanks for your suggestions. Observational evidences show that signals of surface warming exists over fire-prone area, and even extend to downwind areas. The decreased surface albedo over fire prone areas can make the surface absorb more solar radiation and then enhance air temperature through vertical mixing (Fig. R4a). Then, the downwind areas are influenced by warm advection transportation. It is consistent with the abnormal warming signals in Fig. R4b.

The heat released by fire was not treated in the model based on some researches related to the burning characteristics and fire radiative power. For this biomass burning case, during which the burned biomass is winter wheat straw (Li et al., 2016; Huang et al., 2012b). In East China, especially in the northern Jiangsu and Anhui province, most fires of wheat straw are characterized by short-lived smoldering (Fig. R5) (Huang et al., 2012a; 2016). When crops are harvested by hand, the residue is often burned in piles that may smolder, together with short-lived burnings of wheat roots over the surface. The combustion process of smolder is not as full as flaming. Correspondingly, the fire radiative power (FRP) between this straw burning is much weaker than the grassland fire in North America (Fig. R6) based on MODIS Thermal Anomalies Product (MOD14A1 and MYD14A1). The relationship between fire radiative power and fire size varies at a global scale (Laurent et al., 2019). As shown in Fig. R6, the average maximum FRP for the most severe area in this burning case is almost less than $0.02 \text{ kW/m}^2$. As for radiative effects, the maximum of solar shortwave radiation reaching the surface in summer can be over $1 \text{kW/m}^2$, and the sunshine duration is about 11~12 hours (Wallace and Hobbs, 2006). Moreover, surface albedo decline in this place can be over -0.12, as aforementioned (Fig. R2). The energy disturbance aroused by surface albedo decline can be much larger than this kind of straw burning. Therefore, the fire induced warming was not treated in the model.

In addition, the observational signals not only exists on the burning days, but also on the following days (Fig. R7). According to the time series of fire counts (Fig. 1a), the burning had already gone to the end during 16th to 18th June, but surface warming signals still exists over both local and downwind area. The decreased surface albedo can maintain for a period of time before char materials are removed by weathering and new-generated vegetation. But the heat from smoldering cannot last as long. The abnormal warming can be explained by direct radiative effect of decreased surface albedo and the influence of warm advection.

Related description has been added. Please see line 131-137, 219-221 and 280-283 in the revised manuscript.

[Figure]

**Fig. R4 (a)** Temperature bias at 2 m height between FNL data and station observations (OBS), and the 'burned scars' on 13 June 2012 at local time 20:00. Bias is defined as value of OBS minus FNL. Grey arrows mark the 10-m wind field in FNL. Grey dots mark the 'burned scar' (defined as accumulated fire counts in the past 5 days) while orange dots mark fire counts on the day. **(b)** Temperature bias at 2 m height between CTL run and ABD (ALB- △ modisalb) run, and the 10-m wind field in ABD run, on 13 June at 12:00.

[Figure]

**Fig. R5** A photo showing the field burning of wheat straw in Suixi county (33°54′37″N, 116°45′46″E), northern Anhui province on June 14, 2013.

[Figure]

**Fig. R6** MODIS-detect maximum fire radiative power (FRP) during crop straw fire in northern Anhui on 9 June 2012 and grassland fire in North America on 3 July 2004.

[Figure]

**Fig. R7** Temperature bias at 2 m height (fill-coloured circles) between FNL analysis data and station observations (OBS) in zone YHR **(a)** on 16 June 2012, **(b)** on 17 June 2012 and **(c)** on 18 June 2012, at local time 20:00. Bias is defined as value of OBS minus FNL. Grey arrows mark the 10-m wind field in FNL. Grey dots mark the 'burned scar' (defined as accumulated fire counts in the past 5 days) while orange dots mark fire counts on the day.

*Minor Suggestions:*

*Page 1, line 8: impacts air quality*

*Page 1, line 13: show that surface*

*Page 1, line 15: show a positive deviation*

**Response:** Accepted. Great thanks for your carefulness. Please see Line 8, 13 and 15 in the revise manuscript.

*Page 4, line 102: meteorological observations from*

**Response:** Accepted. In addition to make it more readable, the sentence has been rephrased. Please see Line 111-112 in the revise manuscript.

*Page 7, line 221: "and the behind physical images has" to "and the underlying physical images have not"*

**Response:** Accepted. Please see Line 251 in the revise manuscript.

*Page 7, line 222-224: please rephrase this sentence to make it easier to read*

**Response:** Done. To make the whole manuscript more readable, this sentence have been deleted during the modification, and related description has been put in the introduction part. Please see Line 66-68 in the revise manuscript.

*Page 9, line 262: change the sentence to "A typical and severe burning episode in 2012 was chosen as the study case."*

**Response:** Accepted. Please see Line 292 in the revise manuscript.

*Page 9, line 279: naturally induced*

**Response:** Accepted. Please see Line 312 in the revise manuscript.

**References**

[revised manuscript text omitted]